# Bereavement Needs Assessment in Nurses: Elaboration and Content Validation of a Professional Traumatic Grief Scale

**DOI:** 10.3390/ijerph19052968

**Published:** 2022-03-03

**Authors:** Ester Gilart, Isabel Lepiani, María Dueñas, Maria José Cantizano Nuñez, Belen Gutierrez Baena, Anna Bocchino

**Affiliations:** 1University Hospital Jerez de la Frontera, 11407 Cadiz, Spain; esther.gilart@gmail.com; 2Nursing University Salus Infirmorum, 11001 Cadiz, Spain; isabel.lepiani@ca.uca.es (I.L.); belen.gutierrezbaena@ca.uca.es (B.G.B.); anna.bocchino@ca.uca.es (A.B.); 3Department of Statistics and Operational Research, University of Cadiz, 11406 Cadiz, Spain; 4Research Unit, Biomedical Research and Innovation Institute of Cádiz (INIBICA), Puerta del Mar University Hospital, University of Cádiz, 11009 Cadiz, Spain; 5Emergency Department Bahia de Cadiz, La Janda, 11001 Cadiz, Spain; mjcanty14@hotmail.com

**Keywords:** nursing, COVID-19, traumatic grief, scale, validation

## Abstract

The COVID-19 pandemic has caused a series of biopsychosocial repercussions among nursing professionals. The impossibility of anticipating the events, the numerous deaths, the excessive workload, the lack of personal health and the necessary means of protection made it difficult to regulate the impact and the elaboration of grief to the point of becoming, on many occasions, a traumatic grief whose physical and psychological manifestations are becoming more and more evident. The main objective of this research was to develop a scale for a group of symptoms based on professional traumatic grief. The development consisted of two phases: (I) instrument design through a literature review and focus groups of bereavement experts and healthcare professionals who experience the grief process in their work; and (II) validation of the content of the instrument. A total of 25 final items were established as suitable for inclusion in the instrument. It is expected that the experiences and results obtained through the development and validation of a scale of specific symptomatology of professional traumatic grief in health professionals will allow the assessment and detection of symptomatology in order to develop programs and strategies for early intervention and prevention.

## 1. Introduction

The current COVID-19 health emergency that has greatly affected countries around the world could be seen as the result of a complex system of institutional and governmental failings that has caused considerable effects at social, political, cultural and healthcare levels. According to World Health Organization (WHO) data, as of 8 October 2021, 236.8 million people have been infected worldwide. Of these, the death toll is 4.8 million people, thus categorizing this disease as a global emergency [1]. Spain is not exempt since the number of confirmed cases of COVID-19 to date is 4.97 million, with the disease being responsible for 86,701 deaths, and Catalonia, Madrid and Andalusia being the most affected communities [2].

Many researchers [3,4] have wasted no time in conducting studies on the short- and medium-term effects of the pandemic and analyzing the possible impacts and consequences of the COVID-19 crisis on the health of individuals.

One of the most alarming and revealing problems is the suffering experienced by the relatives of COVID-19 victims and healthcare professionals. This has fueled growing interest and more research into the grieving process they experience, which, despite being considered a natural experience in the life of an individual, continues to be a negative experience for most people that is accompanied by serious negative effects on their bio-psychosocial health. Moreover, as is well-documented in the literature, a significant minority of mourners experience a long-term symptomatology that affects their quality of life and well-being and favors the onset of complicated grief, also referred to as prolonged grief disorder (PGD) or persistent complex grief disorder (PCBD). This condition usually manifests itself six months after the loss [5].

Approximately 10% to 15% of people who experience the death of a person will develop complicated grief (CG), and therefore this condition affects tens of millions of people worldwide [6,7]. In this regard, bereavement researchers are concerned that the COVID-19 pandemic is a potential driver of increases in complicated grief [8,9,10,11].

In general terms, complicated grief is characterized by persistent longing or preoccupation for the deceased, symptoms of emotional distress and functional impairment beyond six months after the loss [12]. Regarding the time criteria for diagnosis, the established times vary depending on the circumstances and the subject involved; however, it is estimated that the onset of complicated grief would be after six months of the loss of the deceased person [5]. Given its prevalence and the negative outcomes for health, efficacy and viability, interventions for CG are needed. Therefore, knowing its predictors and specific symptomatology could facilitate the prevention, follow-up and treatment of affected individuals, allowing for a better identification of the diagnosis to be treated. Different authors [13,14] have detailed how personal factors related to the deceased, the disease or circumstances of death and relational aspects play an important role in deterioration after a loss. Another aspect of relevance, especially in the current context, dictated by the COVID-19 pandemic, are the factors related to the disease and the circumstances of death. In fact, sudden and multiple deaths, painful pathological processes and uncontrolled symptoms, mistrust, uncertainty or lack of knowledge about medical treatment are considered predictors of complicated bereavement. The type and volume of losses experienced by the bereaved also affect the grieving process and the likelihood of prolonged bereavement. In this regard, numerous research studies have attempted to predict the circumstances and characteristics of COVID-19-related deaths and how they will lead to increased manifestations of persistent and disabling grief [15,16,17,18]. As a result, the situation becomes more complicated, not only for those individuals or families who have not been able to assist the person in their terminal state, but also in terms of the psychological, medical and economic toll on health professionals’, who have had to face an unprecedented situation [19]. Losses have become established as a special phenomenon, as coping with death has become more frequent, intense and immediate. For bereavement experts, sensitization and subsequent awareness of the distinctive risks of complicated bereavement associated with the pandemic represent pressing issues that need to be addressed [19].

Health teams have witnessed a sharp increase in the frequency with which they have had to deal with the death of not only patients but also colleagues as a result of the virus. This situation has resulted in a series of negative repercussions on the health of professionals, especially nursing professionals, who have provided personalized and comprehensive care to hospitalized patients who have died [20,21,22].

Although the rate of complicated bereavement among healthcare professionals during the pandemic is difficult to estimate due to the short time that has passed since the main waves, there is already extensive literature on the iatrogenic effects of the pandemic. The impossibility of anticipating the events, the multiple sudden deaths of infected patients in a short period of time, the excessive workload of the health team and the lack of personal hygiene and the necessary means of protection have made it difficult to regulate the impact and the elaboration of grief to the point of it often becoming a traumatic grief whose physical, psychological and emotional manifestations have become increasingly evident [23,24]. The effects of complicated bereavement on health are not strictly limited to the physical dimension; rather, they extend to the area of psychosocial health, as mentioned above. The symptomatology includes anxiety, fear and other emotional states, such as feelings of helplessness, insomnia, psychological distress, exhaustion, depressive symptoms, somatization and feelings of stigmatization and frustration [25,26,27,28]. Some manifestations of post-traumatic stress disorder (PTSD) have even been observed in these health professionals [29,30,31]. If we also take into account that these consequences can damage the state of mind of these workers, resulting in lower work performance and a greater risk of errors or omissions, the situation becomes urgent and paramount. Despite its clinical importance, professional traumatic grief was first identified as a specific health problem for the nursing profession only last year, when the term “professional traumatic grief” was presented and validated as a possible diagnostic label [32]. The next step, according to the studies reviewed, should be directed towards the development of specific measurement instruments for identifying the possible risk factors and symptomatology of this disorder in order to plan and take measures aimed at preventing its long-term effects. This would mitigate the threat to personal identity, foster coping resources in professionals, help to minimize the negative self-evaluation of the loss, and improve the quality of life and healthcare of this at-risk population. Although there are instruments suitable for diagnosing prolonged or pathological grief, it seemed appropriate to perform this study to cover certain deficiencies they encounter. Among these, it should be noted that the main instruments found in the literature refer to the symptomatology of the mourner with respect to the loss of a near and dear person [5,33] and/or in specific bereavement situations (e.g., suicide, perinatal death).

Specifically, more general instruments such as the Inventory of Complicated Grief [34], the Revised Texas Grief Inventory [35] or the Grief Evaluation Measure [36], which are the most widely used in the literature, are not adapted to the construct we intend to measure since the items refer to a closer relationship with the deceased.

In addition, specifically for professionals, there are other scales that measure attitudes or skills to cope with death or measure and evaluate their ability to cope with the death of a patient [37], but there is no specific symptomatology scale for health professionals who experience traumatic grief due to multiple patient deaths in a sudden, unexpected and unfair situation.

In other words, considering that the negative consequences of prolonged bereavement can affect the quality of care and the inexistence of a specific traumatic grief scale for healthcare professionals, the present study aims to develop and validate a symptomatology scale specific to “professional traumatic grief” that is clearly defined, easy to understand and generalizable.

## 2. Materials and Methods

This was a mixed study on construction and content validation of a scale for the identification of individuals with a diagnosis of professional traumatic grief.

### 2.1. Phase I: Instrument Design

The instrument was developed through three successive steps that, in accordance with the recommendations established by Polit and Beck [38], follow a qualitative methodology (phases a and b) and a quantitative methodology (phase c). These phases are summarized and grouped as follows: A. Systematic review; B. Focus group of “experts” and professionals experiencing grief; and C. Preliminary design of the inventory.

(A) Systematic review. As a first stage, an initial and exhaustive literature review was necessary. The empirical and theoretical literature is well-suited to providing insights into the phenomenon under study [39]. Systematic reviews, although designed to answer discrete questions using explicit methods, can further contribute to the delineation of the construct [40]. In this regard, the aim of this first phase was to conduct a systematic review of the scientific literature to gather current knowledge about bereavement, complicated grief, bereavement in healthcare professionals and validated instruments in the literature for measuring it. The search strategy for the empirical review was designed to retrieve the largest number of references relevant to measuring professional traumatic grief. The review strategies were guided by standardized approaches, including the Preferred Reporting Items for Systematic Reviews and Meta-analyses (PRISMA) guidelines [41]. The search strategy was configured by combining the following descriptors: “professional bereavement”, “complicated bereavement”, “post-traumatic stress”, “scales”, “assessment”, “health professionals”, “nursing professionals”, “risk factors” and “coping with death”.

(B) Focus group of “experts” and professionals experiencing grief. To encompass a wide range of typologically relevant opinions and observations for the further development of the scale items, focus group sessions were conducted with two samples, namely the expert team and health professionals, with bereavement experiences. Theoretical purposive sampling was used to select participants. The group of experts and the group of professionals experiencing grief were identified by the research group through contacts at meetings with health professionals, membership in related professional organizations and existing professional contacts. An invitation was sent to all potential participants via e-mail. Those interested in participating were contacted by telephone to arrange an appointment, and a few days before the session, participation was confirmed with another telephone call. Participation was voluntary, with the confidentiality and privacy of their contributions ensured. The participants received the necessary information on the objectives, background and methods before providing a written consent form.

The team of experts consisted of professionals with knowledge of bereavement, while the second group consisted of professionals who experience grief (these professionals are nurses, doctors, healthcare assistants and other healthcare professionals who work in services where death is most present (ICU, emergency, oncology, pediatric oncology, etc.). Given the nature of the objectives of the study, the formation of focus groups was considered the most relevant method for obtaining information and gaining a more in-depth understanding of the topic in order to construct a questionnaire aimed at a sample of individuals with professional traumatic grief. The focus group meetings were conducted by researchers from the team trained in qualitative research and were held in a comfortable, quiet place to ensure quality communication and preserve confidentiality. The place chosen was the meeting room of the Salus Infirmorum University Nursing Center.

At the beginning of the session, all the participants were provided with a definition of the construct to be measured, i.e., professional traumatic grief, and were asked to express their opinions and perceptions regarding the subject matter, the characteristics of prolonged grief and its possible inclusion in a symptomatology scale. In the case of health professionals, their experiences of patient deaths were also discussed. Prior to the focus group meeting with the health professionals, the researcher ensured that the participants felt comfortable being interviewed and, after the interview, they were offered emotional support if needed. The discussion was facilitated by some open-ended questions to generate a voluntary debate on the different perspectives of the professionals about their experiences of patient deaths. The information obtained was audio-recorded so it could be transcribed verbatim and then analyzed as detailed by Krippendorff [42], using conventional methods of qualitative content analysis. The moderators used a semi-structured approach and an interview guide. The interview guide was constructed to specifically explore concepts relevant to professional traumatic grief and its components. A field notebook was also used to collect nonverbal aspects of communication (gestures, expressions, postures, etc.) in order to obtain the results of what was expressed by the participants.

Data analysis: Focus group analysis was understood to be the process of the identification, codification and categorization of the main axes of meaning underlying the data. Qualitative data from the focus group were transcribed and processed.

For their correct analysis, the steps of Giorgi’s phenomenological method [43] were followed:(1)Collect and describe the phenomenological data.(2)Read the description of the data set.(3)Divide the descriptions into units of meaning.(4)Transform the units of meaning.(5)Identify the essential structure of the phenomenon.(6)Integrate the characteristics into the essential structure of the phenomenon.

Specifically, each researcher listened to the recordings several times to obtain a comprehensive understanding of the phenomenon under study. After transcription and subsequent reading, the units of meaning (words, constructs and/or concepts) related to the construct were labeled with a code. Similar codes representing similar concepts were classified into subcategories and then converted into a category. Data collected from health professionals’ focus group and from experts’ focus group were analyzed and reported separately and refined and reduced until mutual agreement between the researchers was reached.

(C) Preliminary Instrument Design. In this phase, a first version of the instrument was developed, consisting of 28 items reflecting the symptomatology of professional traumatic grief. The questionnaire items were designed based on the symptoms of prolonged grief most frequently reported in the scientific literature [5,25,26,27,28] on published instruments of grief [27,28,29] and on the information obtained from the focus groups. In addition, it was considered appropriate to include among the symptoms the defining characteristics that obtained high scores in the article by Gilart et al. [32].

The research team named the scale Inventory of Symptoms of Professional Traumatic Grief, or ISDUTYP.

A 7-point Likert-type response scale was chosen, where 1 indicates never, 2 hardly ever, 3 seldom, 4 sometimes, 5 often, 6 usually and 7 always. In addition, the experts agreed that with respect to the wording of the instrument, the participants had to indicate the frequency with which they felt the symptoms during the last six months.

### 2.2. Phase II: Validation of the Content of the Instrument

Content validity is defined as the degree to which the items of an assessment instrument are relevant and representative of the entity to be measured. This method is characterized by having a number of experts who either propose the items or dimensions that should make up the construct of interest or evaluate the different items according to their clarity, coherence and relevance, based on a Likert-type scale, and make judgments on the elements and contents to be evaluated [44]. Evaluating the content validity of a scale is an essential step in the development of an instrument, since doing so allows the appropriate items for the scale to measure to be selected. The procedure recommended by Polit and Beck [38] to ensure the content validity of an instrument is based on expert judgment of the relevance of the items to the construct they are intended to measure. The main objective of the second phase was to submit the indicators of the version elaborated in the previous phase to the judgment of a group of experts who would analyze the clarity, coherence and relevance of each item, taking as a reference the classic criteria established by Angleitner, John and Löhr [45]. These criteria include: level of clarity (the item is well-understood and its syntax and semantics are adequate); coherence (the item has a logical relationship with the scale of professional traumatic grief); relevance (the item is important and should be included in the scale). In addition, at the end of the questionnaire, there was a text box where it was possible for the judges to write the comments and suggestions they considered pertinent.

### 2.3. Evaluation of the Clarity, Consistency and Relevance of the Symptomatology Scale by a Panel of Experts

For the selection of experts, there is no consensus to define what constitutes an expert, but it is important that they have knowledge in the area under investigation and that they work on an academic and/or professional level and, in turn, have knowledge of complementary areas [46]. The number of experts selected was determined according to authors’ recommendations for obtaining useful estimates, the ideal number being between 7 and 30 experts [47,48].

The hyperlink that was generated to evaluate the instrument was sent by e-mail to:-College of Nursing in the province of Cadiz;-Experts on the subject, who were accessed through: direct contact; bereavement research groups; nursing faculties in the province of Cadiz. To conduct this phase, the QuestionPro platform was used to prepare and digitize a document that included:-Data about sociodemographic characteristics: sex, age, years of experience, position held (nurse, doctor, psychologist, teacher, researcher, other), unit/service where they worked (hospital, primary care, university, emergency services, other);-The definition of professional traumatic grief according to Gilart et al. [32] and the symptomatology to be evaluated with a questionnaire to assess it.

The evaluation focused on assessing the clarity, consistency and relevance of each of the symptoms included in the questionnaire. A four-point ordinal scale (1 = not clear, 2 = somewhat clear, 3 = quite clear and 4 = very clear) was used to assess the clarity of each of the symptoms, and a four-point ordinal scale (1 = not coherent (or relevant), 2 = somewhat coherent (or relevant), 3 = quite coherent (or relevant) and 4 = very coherent (or relevant)) was used to assess the coherence and relevance of each of the symptoms. Finally, for each of the sections to be evaluated by the experts, a space for open answers was added for new suggestions or reformulating symptoms that were difficult to understand.

The statistical analysis of the data generated from the experts’ responses included a descriptive analysis of the sociodemographic variables and the calculation of item- and scale-level content validity indices (I-CVI and S-CVI, respectively). For each item, the I-CVI was calculated, defined as the proportion of experts who rated the content as valid (representativeness/clarity rating of 3 or 4) [49]. Items were rated as excellent when the I-CVI value was greater than 0.78 [50]. For full validation of the scale, all I-CVI values were averaged to calculate the S-CVI, for which a value above 0.90 was considered excellent [49]. The sociodemographic data were analyzed with SPSS for Windows version 23 (IBM, Armonk, NY, USA) and the I-CVI and S-CVI with Microsoft Office Excel, 2016 (Microsoft Corporation, Redmond, WA, USA).

## 3. Results

### 3.1. Phase I: Instrument Design

(A) Systematic review. After a review of the literature, 25 articles were found on instruments focused on the assessment/measurement of complicated grief symptoms, with the most commonly used scales being the Complicated Grief Inventory [34], the Texas Revised Grief Inventory [35] and the Grief Assessment Measure [36]. The most recent validation scale turned out to be The Traumatic Grief Inventory-Self Report Plus (TGI-SR+), which also had a good reliability index [51] However, not all articles fully met the eligibility criteria as no results were shown for a prolonged bereavement instrument specific to health professionals.

(B) Design Focus group of “experts” and professionals experiencing grief. A single separate session (one for experts and one for healthcare professionals) took place, lasting approximately 50 min, since the content and quality of the information reached a good level from the first opinions.

A total of 25 people participated in the focus group (15 people for the specific-care-unit nurse group and 10 bereavement experts). The sociodemographic characteristics of the grief experts and professionals experiencing grief are shown in Table 1 (Table 1).

The criterion of heterogeneity was respected both by the group of experts as well as by the group of professionals who have experienced grief.

To ensure the quality and validity of the study (in terms of credibility, auditability and transferability), different strategies were used according to the reliability criteria of Guba and Lincoln [52].

The findings of qualitative data from health professional focus groups are presented below by six themes and illustrated by some quotations from the interviewees (I).

### 3.2. Subjective Experience of Bereavement

Bereavement is experienced by health professionals as something “natural”. From an emotional point of view, it involves, for most participants, a significant but unavoidable loss: loss of the possibility of caring, loss of continuity of life or loss of control over the patient’s life. These losses do not affect the professionals of different care units in the same way, as intensive care nurses seemed to be more “used to it”.
*I1: “You always expect things to get better, but it’s like deep down you expect it”.**I4: “Death is a natural thing, a universal experience, but of course it still hurts”.**I7: “At the end of the day you get used to it… or maybe it’s your coping strategies”.**I15: “at home we don’t talk about it anymore, we used to… I remember that at the beginning we even cried… now the one who cries is the father, the mother or the wife…”.*

### 3.3. Lived Experience of Death from COVID

Participants showed a consensus in considering the death of a patient during the pandemic as something different. The participants’ perception of being treated unfairly by the institutions during the pandemic contributed to an increase in emotional discomfort and mistrust in the healthcare organization.
*I2: “what is most shocking is the reaction of the relatives, it is incredible how we worry more about those who stay than about those who leave”.**I3: “The healthcare system does not train us, does not support us and leaves us alone”.**I4: “The death of a patient from COVID? you are left there, not knowing what to do, and now you have to communicate it to a family member who has not even been able to say goodbye”.**I5: “When a patient dies from COVID, there is a whole series of reactions, feelings and opinions that you don’t even know how to explain, it is something new but at the same time it is so common”.**I6: “I should get used to it just because so many people have died from the same thing? I’m sorry I can’t, it still shocks me”**I15: “It is not the same when a patient dies in palliative care or for other reasons… the situation is very different… and what hurts the most? That the patient is dying alone and you are the professional, friend and family member…”*

### 3.4. Reactions and Feelings after a Patient’s Death from COVID-19

Participants described the reactions and emotions they experienced during the pandemic. There was no single consensus regarding the range of emotions tested; however, many of them reported feeling distressed, sad, helpless and sometimes angry. Altered behavior manifesting as insomnia or tranquilizer abuse was evident in their narratives. Work overload was present in most of the participants, but they did not attribute their emotional distress to this.
*I1: “You feel sad, desolate and disoriented, but above all you feel helpless”.**I8: And where do we leave all that rage, anger? Against the health system, against everyone and maybe even against God”.**I9: “At the beginning I couldn’t even sleep… this feeling of suffocation that came over me every night was horrible”.**I11: “I had quit smoking and maybe because of stress or anxiety in this situation I started again”.**I14: “I feel dissatisfied… with everything… with my work, my family, but above all with myself”.**I15: “We have had a lot of work overload… and we are not burned out… but it is not this… it is sadness, impotence, anger and perhaps the non-acceptance of what was happening”.*

### 3.5. Individual, Family and Social Consequences

The COVID-19 pandemic has brought with it a series of family, social and occupational repercussions for health professionals. Most of the informants reported social isolation; on the one hand, this isolation was voluntary, i.e., they themselves decided to withdraw from their social group due to fear of contagion or as a result of emotional distress, and on the other hand, they felt excluded by their group of closest friends and relatives.
*I1: “I isolated myself… completely… yes… I isolated myself, I was afraid”.**I2: “I distanced myself from my family and my group of friends for fear of contagion… well, contagion”.**I6: “I was not well, I preferred to be alone, but they didn’t make it easy for me either, I could feel how people were pushing me away”.**I7: “I asked a friend to pick up my child at school… and she said no, that it would have been better to avoid it for a while”.**I9: “I was bullied… yes, I was bullied… she didn’t want me to hang out with anyone in the gang… I wouldn’t have had the time, but being told that it was ugly”.**I14: “Even today, now that the situation is calmer… every time I go into the ICU I have flashbacks and sometimes I feel anguish and anxiety”.*

### 3.6. Family and Organizational Support for Healthcare Professionals

Participants described the importance of social and family support during the pandemic. Several spoke of increasing inner turmoil, a sense of sadness that only family members could alleviate.

They especially emphasized the importance of feeling supported by family members and coworkers.
*I2: “The only thing that mattered to me when I came home was to see my family”.**I3: “When I came home from work, the first thing I thought was: I hope it never happens to me or my loved ones”**I4: “I remember once, when I came home from work, the first thing I thought was: I hope it never happens to me or my loved ones”.**I5: “I remember one time, I think it was in the second wave… I came home, took a shower, lay down on the bed and started crying like a little girl… my little daughter came and all she did was hug me… it was nice and refreshing”….**I7: “My husband always supported me throughout the whole process… there were even times when he would argue with my neighbors for making me feel “a danger” to the community…”**I9: “The support of your loved ones is important… as well as your coworkers or even your boss”.*

### 3.7. The Importance of a Scale for Diagnosing Professional Traumatic Grief

The theme “the importance of a scale to diagnose professional traumatic grief” encompasses the ability to organize and prioritize the nurse’s needs in order to guide the development of necessary interventions for their correct recovery. All the participants considered it necessary to create and validate a diagnostic scale to identify the feelings and emotions experienced during the pandemic. Many of them underlined that recovery is important in promoting quality and efficient healthcare.
*I1: “I don’t know if it’s grief or something else… but there is something wrong with us… there are days when I can’t get my head up… and we have to find out”.**I4: “it is essential to know what has happened to us and what is still happening to us… They want us to do our job well? Then we have to find solutions”.**I7: “I would be happy to receive any kind of help”.**I8: “These two years have been terrible, it’s time to get back on my feet and I can’t do it alone”!**I14: “Early diagnosis, it has always been said that it is essential… this time it is time for self-diagnosis because something is wrong”.**I15: “I wouldn’t mind going to therapy or taking 20 questionnaires to find out what is happening to me”.*

In turn, the information obtained from expert focus groups was composed of supposedly homogeneous opinions. The participants emphasized the need for and importance of developing a specific instrument for professional traumatic grief. Moreover, considering the possibility of misdiagnosing bereavement during the pandemic, they underlined that a questionnaire of symptomatology could be a first and necessary step for the early detection of a real problem that health professionals are experiencing.


*I2: “It is no longer just a matter of depression or anxiety, they have suffered a lot and professionals have the right to be acknowledged that they have experienced grief in their work”.*

*I5:” a specific instrument for professional traumatic grief is a priority”*


(C) Preliminary Instrument Design. In this phase, in addition to the items to be included in the scale and their response alternatives, the structure of the scale was defined, as well as the instructions and the response format of the instrument. With respect to the results of the different aspects to be addressed, the following was decided:

The instrument would consist of 28 items reflecting the symptomatology of professional traumatic grief.

### 3.8. Phase II: Validation of the Content of the Instrument

The content validity analysis was performed with the participation of the 30 expert judges. The sociodemographic characteristics of the selected experts are shown in Table 2.

Of the 30 participants, 76.7% were women, with a mean age of 44.17 years (SD = 8.498). Most had an academic level of nursing, 40.0% were nurses, 36.7 physicians, 16.7 psychologists, 3.3 teaching staff and 3.3 researchers. On average, they had 18.17 (SD = 8.844) years of experience in the field, 30.0% in hospital, 30.0% in primary care, 6.7% in a university, 26.7% in emergency and health emergency services and 6.7% in other contexts.

The results of the ISDUTYP validation process are described below in Table 3, which shows the I-CVI of each item of the scale to assess clarity, coherence and relevance. In the clarity section, most of the items presented values above 0.78, except items 12 and 19. Regarding the coherence of the items, all of them obtained excellent values, all scoring higher than 0.80. Regarding relevance, items 12, 19 and 25 obtained values lower than 0.75, so they were eliminated from the inventory. Finally, an S-CVI value of 0.90 was obtained for clarity, 0.88 for coherence and 0.91 for relevance.

Based on the modifications made after the review by the group of 30 experts, the final version of the ISDUTYP was obtained with a total of 25 items.

## 4. Discussion

With regard to the results obtained in the different phases of the study, it is worth highlighting the following:

1. The review of the literature showed a lack of specific instruments for an accurate diagnosis of “professional traumatic grief”. This methodological gap is probably due to the fact that the most widespread theories when dealing with loss and grief are related to the emotional response of a subject to the loss of a strong emotional bond [53] and it has rarely been considered as persistent suffering of the mourner regardless of their relationship with the deceased, and even less so in a work context. Furthermore, despite the existence of numerous scales for measuring pathological grief, their specificity makes them difficult to use in some fields of research, so much so that many studies opt for a qualitative methodology [5,33]. In our opinion, although deepening the understanding of aspects of such a complex construct as prolonged bereavement is worthy of attention, it is not possible to achieve this without reliable and rapid instruments that can reduce the time required for early detection and consequent interventions. This is particularly relevant when referring to professionals who are so valuable in a situation of public emergency. This finding can be reflected in the discussion groups. Both groups stressed the importance of developing a new instrument to specifically measure professional traumatic grief. In fact, the participants themselves reported that most of the time they cannot identify what has happened to them and do not know how to cope with it.

2. For the content validation of the instrument, the theoretical criteria described in the literature were considered by means of expert judgment and the consequent calculation of the content validity index [38]. The final instrument had a total of 25 items.

The results support the argument that the ISDUTYP exhibited excellent content validity for clarity (0.90) and relevance (0.91). For consistency, the results show an acceptable I-CVI (0.88).

Most of the proposed symptomatology items achieved good scores, with each item being considered clear, relevant and coherent regarding the construct to be measured.

In fact, according to the literature on clinical symptomatology and the diagnostic criteria for prolonged bereavement [5,25,26,27,28], the items that obtained the highest scores in terms of relevance were sadness, emotional exhaustion, continuous negative mood, despair, depression and adjustment problems. The same symptoms achieved good LCI scores with respect to the criteria of clarity and coherence.

Still focusing on relevance, the symptoms that obtained the lowest I-CVI were item 13 (re-experiencing), item 17 (experience of repetitive unpleasant memories or images) and item 18 (avoidance of places or objects or thoughts reminiscent of the event). However, there was extensive reference literature linking prolonged grief to PTSD [54]. There is also inconsistency among different researchers in considering them as independent disorders or as aspects of the same pathology, a fact that is also reflected in our results. Something similar can be observed in item 22 (panic attack) and item 11 (feeling of irritability), since there seems to be a discordance when it comes to establishing a greater relevance for their inclusion in the instrument. However, it should be noted that none of these items were excluded since their I-CVI was considered unacceptable.

The symptoms that were not considered clear were modified to facilitate their understanding; however, if the same symptom was considered unclear or coherent and also not very relevant, it was automatically excluded from the final scale. Following this same logic, item 12 (gastrointestinal problems), item 19 (somatizations) and item 25 (weakness, dizziness and vertigo) obtained low scores for clarity and relevance and were therefore not included in the final version of the instrument. This initially caught our attention due to the various research studies on the physical and biological consequences that complicated bereavement can have [55].

However, in accordance with the first results obtained in the focus groups, a possible explanation could be related to the factors that are supposed to predispose individuals to prolonged grief disorder. That is, in cases such as prolonged grief related to the victims of COVID-19, grief processes are much more closely related to psychological factors than physical factors. Moreover, accumulated and/or insufficiently elaborated grief and stressful life events often result in mental imbalance, although the accumulation of intervening variables makes it difficult to find causal relationships. In our opinion, after reading some of the suggestions in the text box, physical symptoms were considered immediate manifestations of a significant loss, leaving room for psychosocial symptomatology or behavioral disorders to manifest themselves with time (in the case of prolonged bereavement).

### Limitations

The use of a sample of experts in this study gives rise to a number of limitations with respect to the subjectivity and generalizability of the results. Since expert feedback is subjective, the study was subject to possible biases. Another aspect to take into account when interpreting and applying the results is that their generalizability to international populations may be limited. The development of the questionnaire based on the cultural characteristics of a single country may lead to the need for future cross-cultural adaptations of certain symptomatology to other contexts.

## 5. Conclusions

The present study has introduced the development of a specific measurement instrument suitable to identify possible risk factors and a symptomatology of professional traumatic grief. It could be used to plan and take action aimed at preventing the long-term effects of this pathology, thereby mitigating the threat to personal identity, promoting coping resources for professionals, helping to minimize negative self-evaluation due to the loss and improving the quality of life and healthcare of this at-risk population.

Finally, it should be noted that the next step is oriented to the analysis of the psychometric characteristics of the ISDUTYP in a sample of nurses who have worked on a daily basis during the pandemic. In addition, it is believed that the final version of the instrument will be useful in other bereavement contexts and for other health professionals. Although the choice of post-COVID professional traumatic bereavement is dictated by the current frame of reference, its use is intended for experiences in other situations in which professional bereavement is a risk (specific multiple death units, e.g., oncology, intensive care units, etc.) and/or even by other professions (firefighters, police officers, etc.).

## Figures and Tables

**Table 1 ijerph-19-02968-t001:** The sociodemographic characteristics of the grief experts and professionals experiencing grief.

Group of Experts			
Variables	N	Mean (SD)	Percentage
Sex			10
Male Female	3 7		30% 70%
Age		46.40 (7.367)	
Length of experience in position		17.40 (8.669)	
Position in the institution			10
Nurse	3		30%
Physicians	3		30%
Psychologists	2		20%
Teaching staff	1		10%
Researchers	1		10%
Unit/Service where you work			10
Hospital	3		30%
Primary care	2		20%
University	3		30%
Emergency and Health emergency Services	2		20%
**Group of Professionals Experiencing Grief**			
Sex			15
Male Female	4 11		26.67% 73.33%
Age		44.47 (6.802)	
Length of experience in position		20.80 (7.113)	
Position in the institution			
Nurse	15		
Unit/Service where you work			15
Intensive care unit (ICU)	5		33.33%
Pediatric intensive care unit	3		20%
Emergency	2		13.33%
Oncology	2		13.33%
Pediatric oncology	3		20%

**Table 2 ijerph-19-02968-t002:** Sociodemographic characteristics of the selected experts.

Variables	N	Mean (SD)	Percentage
Sex			
Male Female	7 23		23.3 76.7
Age		44.17 (8.498)	
Length of experience in position		18.17 (8.844)	
Position in the institution			
Nurse	12		40.0
Physicians	11		36.7
Psychologists	5		16.7
Teaching staff	1		3.3
Researchers	1		3.3
Unit/Service where you work			
Hospital	9		30.0
Primary care	9		30.0
University	2		6.7
Emergency and health emergency Services	8		26.7
Others	2		6.7

**Table 3 ijerph-19-02968-t003:** Content validity index (I-CVI).

Item	Clarity		Coherence		Relevance	
	I-CVI	Experts Agreeing	I-CVI	Experts Agreeing	I-CVI	Experts Agreeing
1. Feelings of sadness	0.97	29/30	0.93	28/30	0.97	29/30
2. Feelings of helplessness	0.97	29/30	0.97	29/30	0.87	26/30
3. Feelings of fear	0.87	26/30	0.90	27/30	0.90	27/30
4. Emotional exhaustion related to the traumatic event	0.97	29/30	0.93	28/30	0.97	29/30
5. Difficulty falling or staying asleep	0.90	27/30	0.90	27/30	0.97	29/30
6. Anxiety	1	30/30	0.97	29/30	0.93	28/30
7. Guilt about the causes or consequences of the traumatic event	0.83	25/30	0.87	26/30	0.87	26/30
8. Continuous negative mood in the form of rage, anger and/or annoyance	0.97	29/30	0.93	28/30	0.97	29/30
9. Desperation	0.93	28/30	0.93	28/30	0.97	29/30
10. Memories, thoughts, or feelings of distress	0.90	27/30	0.83	25/30	0.90	27/30
11. Feelings of irritability	0.87	26/30	0.87	26/30	0.83	25/30
12. Gastrointestinal problems (nausea, diarrhea, others)	0.77	23/30	0.80	24/30	0.73	22/30
13. Performing behaviors or experiencing sensations or emotions as if the event were happening again	0.87	26/30	0.83	25/30	0.90	27/30
14. Experiencing unhealthy behaviors (smoking, drinking, others) after the traumatic event	0.83	25/30	0.83	25/30	0.93	28/30
15. Depression	0.97	29/30	0.93	28/30	0.97	29/30
16. Feeling of dullness	0.97	29/30	0.90	27/30	0.90	27/30
17. Experiencing unpleasant and repetitive memories or images of the event involuntarily	0.87	26/30	0.87	26/30	0.83	25/30
18. Avoidance of places, objects or thoughts that recalled the event	0.80	24/30	0.87	26/30	0.83	25/30
19. Appearance of physical symptoms as a response to psychological factors	0.70	21/30	0.83	25/30	0.70	21/30
20. Difficulty breathing	0.80	24/30	0.80	24/30	0.90	27/30
21. Intrusive thoughts or images related to the traumatic event	0.87	26/30	0.83	25/30	0.93	28/30
22. Panic attacks	0.90	27/30	0.83	25/30	0.80	24/30
23. Loss of ability to care about patients or feeling distant with patients you used to care about since the traumatic event	0.93	28/30	0.90	27/30	0.93	28/30
24. Difficulty in accepting reality	0.97	29/30	0.93	28/30	0.93	28/30
25. Weakness, dizziness and/or lightheadedness	0.80	24/30	0.83	25/30	0.70	21/30
26. Experiencing confusion when remembering what happened	0.87	26/30	0.83	25/30	0.90	27/30
27. Nervousness or tremors	0.80	24/30	0.83	25/30	0.80	24/30
28. Problems adapting to the situation experienced	0.97	29/30	0.93	28/30	0.97	29/30

## Data Availability

The data presented in this study are available on request from the corresponding author.

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
