# Peer review of "Bereavement Needs Assessment in Nurses: Elaboration and Content Validation of a Professional Traumatic Grief Scale"

_ijerph, 2022, doi:10.3390/ijerph19052968_

Round 1

Reviewer 1 Report

It is very valuable article as it focuses on the development of new  tool. 

It would be good to highlight the direction of future / follow up studies. 

In the methodology part , it could be explained why (lines 241-242) you took the number of experts, not only mentioning that according to Varela's recommendations. 

And wider characteristics of experts should be provided. 

Author Response

Dear reviewer, we thank you in advance for all the contributions and sound advice given to improve the article. 

Reviewer 2 Report

I read with interest the article entitled "Bereavement needs assessment in nurses. Elaboration and content validation of a Professional Traumatic Grief scale ». I will definitely agree with the authors that it is necessary to construct a scale that records the Professional Traumatic Grief.

The researchers did a considerable amount of work on the construction of the scale without unfortunately completing the construction of the scale. They constructed the scale in a very serious way and examined the validity of the scale, but a scale can not be useful if its reliability is not examined.

I would ask the researchers to withdraw the paper, to examine the reliability of the scale and then to publish the results.

Finally, let me note a rather typographical error (?) In the paragraph "Conflicts of Interest" line 545-548: We attest to the fact that all the authors… .. and agree to its submission to the International Journal of Nursing Knowledge.

Author Response

Dear reviewer, we thank you in advance for all the contributions and the interesting advice given to improve the article. 

Reviewer 3 Report

Thank you for the opportunity to review the interesting topic “Bereavement needs assessment in nurses. Elaboration and content validation of a Professional Traumatic Grief scale”.

The following observations are made:

  1. The Abstract is not prepared in accordance with the requirements and must be adjusted.
  2. Subsection titles must be corrected by declining uppercase letters.
  3. Lines 274-289: it looks like it's Methods, but it's not Results.
  4. Lines 296-410: information on the qualitative study methods used should be noted by the authors in the Methods section.
  5. The Authors must also consider and identify four ethical issues related to the interview process: reducing the risk of unanticipated harm; protecting the interviewee's information; effectively informing interviewees about the nature of the study; and reducing the risk of exploitation. This information should be written into the section of Methods.
  6. Lines 408-410: Again, the Methods are described, not the Results.
  7. Table 3 is incomprehensible. It seems necessary either to write precise statement of questions (items) or give these items as the supplementary files. It is impossible to understand what the results are specifically about.
  8. Lines 434-436: It remains unclear why the Discussion begins with the aim of the study.
  9. Lines  437-440: This is the Methods recorded in the Discussion section.
  10. Lines  443-462: These are Results, but are listed in the Discussion section.
  11. It must also be recognized that many questions are complex and authors should perform a thoughtful analysis of all the possible methods that can be used to answer a research question. Increasingly, mixed methods in which both qualitative and quantitative approaches are integrated are needed to contribute to a rich and comprehensive study.
  12. Nevertheless, the manuscript has significant limitations regarding the generalization of results (lines: 504-506). At present, the paper does not meet the basic requirements for its preparation (for the Abstract, Methods, Results, Discussions and Conclusions). Information of the study needs to be systematized. The approvement from Institutional Review Board Statement must be claimed too.
  13. Despite the fact that the idea of authors is relevant in the current period, the manuscript has very serious shortcomings.
  14. In addition, the references were not drafted correctly.

Best Regards

Author Response

(The authors gave the same response as above.)

Reviewer 4 Report

Dear Sir/Mam

Please find bellow the requested review regarding the manuscript. The article contains a lot of useful information on the issue. The topic is very interesting and but use of sources is not appropriate. Although it has some useful information there are less references and the statements are not established. I suggest the authors to write more information with references.

The article contains a lot of useful information on the issue. It is quite clear what is already known about this topic and the research question is clearly outlined. There is no structure at the manuscript. The research question is not justified clearly, given what is already known about the topic and it is questionable if there are opportunities to inform future research. Positive: There are some strengths of the article that could have an impact in the field, such as the topic and its impact on the existed literature. The manuscript is approved publication only after major changes.

Author Response

(The authors gave the same response as above.)

Reviewer 5 Report

In what otherwise is an exemplorary study, and the findings support the conclusions, but without identiying Items 1 - 25, there appears to be little information of value to policy makers and those proposed to take remedial action.  The report purportes to "develop and validate a symptomatology scale specific to "professional traumatic grief"". While the research has accomplished this, the results are known only to the authors and accordingly are of no value to policy makers or health professionals. Is this intentional or just an editorial oversight?

Author Response

(The authors gave the same response as above.)

Round 2

Reviewer 2 Report

First of all, I would like to thank the authors for the good news that they have completed the reliability check of the scale. I understand the authors' need to produce as many publications as possible from a research. However this should not be at the expense of the quality of the publications. I think I can imagine future writers splitting the article into not two but several parts with the indication "To be continued". Finally I would like to inform authors that this journal has no page limit therefore I would ask authors to add the reliability check.

Reviewer 3 Report

Thank you for the opportunity to review the topic „Bereavement needs assessment in nurses. Elaboration and content validation of a Professional Traumatic Grief scale“.

In my opinion, the authors responded to my comments, overwhelmingly corrected the manuscript, and provided additional information in the annexes. I recommend that the paper be accepted after minor corrections have been made:

Authors are recommended to use the same character throughout the manuscript (e.g. line 429 “sd” and line 431 “SD”).

Lines 518-522: This information is certainly not a conclusion. In my opinion, this information is unnecessary.

Kind Regards

Author Response

Dear reviewer, first of all, thank you very much for your time and suggestions, which have undoubtedly improved the quality of the manuscript. Secondly, we have proceeded to delete lines 518-522, as you suggested, with their respective references. And finally, we have revised the character throughout the manuscript. Thank you for everything! Best regards 

Reviewer 4 Report

Congrats

Author Response

Dear reviewer,  thank you very much for your time and suggestions, which have undoubtedly improved the quality of the manuscript. Thank you for everything! Best regards 

Reviewer 5 Report

I have to admit I dont understand the reluctance to present the Items in the published article, as findings from the research. The statistical information is interesting, but if no reommendations, what is the purpose?

Author Response

Dear reviewer, thank you again for your time.

For our part there is no reluctance to publish the symptomatology scale, simply, reviewing some articles, published by the same journal, we realized that many authors published the final version of the instrument in supplementary material.  We are sorry for the confusion and have proceeded to modify it as soon as possible by including the symptomatology scale in table 3. We fully agree with you on the fact that without the final version of the instrument it would be impossible to understand the respective conclusions where we include also the recommendations and future suggestions (lines 518-531). Thank you.